# Influence of electric double layer rigidity on CO adsorption and electroreduction rate

Jiajie Hou[1], Bingjun Xu [2] ✉ & Qi Lu [1] ✉

Understanding the structure of the electric double layer (EDL) is critical for designing efficient electrocatalytic processes. However, the interplay between reactant adsorbates and the concentrated ionic species within the EDL remains an aspect that has yet to be fully explored. In the present study, we employ electrochemical CO reduction on Cu as a model reaction to reveal the significant impact of EDL structure on CO adsorption. By altering the sequence of applying negative potential and elevating CO pressure, we discern two distinct EDL structures with varying cation density and CO coverage. Our findings demonstrate that the EDL comprising densely packed cations substantially hinders CO adsorption on the Cu as opposed to the EDL containing less compact cations. These two different EDL structures remained stable over the course of our experiments, despite their identical initial and final conditions, suggesting an insurmountable kinetic barrier present in between. Moreover, we show that the size and identity of cations play decisive roles in determining the properties of the EDL in CO electroreduction on Cu. This study presents a refined adaptation of the classical Gouy-Chapman-Stern model and highlights its catalytic importance, which bridges the mechanistic gap between the EDL structure and cathodic reactions.

With the increasing supply of renewable electricity, electrolysis is likely to play a central role in the future energy and chemical industries[1–5]. Electrocatalytic reactions proceed at electrified interfaces, within which reactants, ions and electrons converge, enabling electrochemical transformations. A key feature of the electrified interface, or the electric double layer (EDL), is the presence of the strong interfacial electric field (-10⁷ V/cm) that dominates interactions among interfacial species and drives the electrocatalytic processes[6–10]. It has been established that the applied potential can significantly influence adsorbed reactive species within the EDL and profoundly affect electrocatalytic processes, as evidenced by recent theoretical works[11,12]. According to the classic Gouy-Chapman-Stern (GCS) model, the composition and structure of the EDL are drastically different from the bulk electrolyte, and thus the EDL and the bulk electrolyte could be considered as two distinct phases, with their boundary delineated by the diffuse layer[13–15]. The GCS model implicitly presumes that the species in the EDL are in equilibrium with the bulk electrolyte in the

absence of any Faradaic processes. This assumption, though widely accepted, is rarely examined.

The electrochemical CO reduction reaction (CORR) is a key step in the electrochemical CO₂ reduction reaction (CO₂RR)[16–19]. Consequently, understanding its reaction mechanism and discerning the factors that impact its rate and product distribution has become a focus of recent literature[20–24]. Cu is the only metal surface with appreciable selectivity for valuable hydrocarbons and oxygenates in the CO₍₂₎RR[25–28]. and the coverage and binding energy of surface-adsorbed CO ($CO_{ad}$) have been proposed to play pivotal roles in determining the CO₍₂₎RR performance[29–33]. Thus, employing $CO_{ad}$ on the Cu surface as a reporter to investigate the structure of the EDL is of both academic and practical importance. We recently determined the standard adsorption enthalpy of CO on Cu surfaces under electrochemical conditions (-1.5 kJ/mol)[34]. and found that CO adsorbs relatively weakly, making $CO_{ad}$ an apt and sensitive probe to the EDL structure. Furthermore, given $CO_{ad}$'s role as a key intermediate in

[1]State Key Laboratory of Chemical Engineering, Department of Chemical Engineering, Tsinghua University, 100084 Beijing, China. [2]College of Chemistry and Molecular Engineering, Peking University, 100871 Beijing, China. ✉e-mail: b_xu@pku.edu.cn; luqicheme@mail.tsinghua.edu.cn

$CO_{(2)}RR$, this approach offers a promising avenue to correlate the EDL structure with catalytic efficacy.

In this work, we combine high-pressure in-situ surface-enhanced spectroscopy and the CORR reactivity measurements to demonstrate that the adsorption of $CO_{ad}$ can be profoundly influenced by the structure of EDL, which is dependent on the order of applying negative electrode potential and elevating CO pressure. Specifically, the EDL established when the negative potential (e.g., −0.9 V) is applied at low CO pressure (1 atm) followed by elevating the CO pressure to 40 barg, denoted as $EDL_{lp}$, is more compact and rigid than the EDL formed when the negative potential is applied post the increase in CO pressure, denoted as $EDL_{hp}$. The $EDL_{lp}$ is identified as a kinetically trapped state with a lower CORR activity. Increase in the chemical potential of CO by elevating its pressure is insufficient to overcome the kinetic barrier associated with the conversion of the $EDL_{lp}$ phase to the $EDL_{hp}$ phase. The structure of EDL is shown to depend on the size of the hydrated cations, which could impact the reactivity via the strength of the interfacial electric field strength and the rigidity of EDL. To the best of the authors' knowledge, this study provides the first experimental evidence for a non-equilibrated EDL structure and its impact on electrocatalytic performance.

## Results

### Suppression of CO adsorption by negative potential at elevated CO pressure

The adsorption of CO at elevated pressures on polycrystalline Cu was investigated with SEIRAS at a typical CORR potential of −0.9 V in a potassium phosphate buffer electrolyte (pH = 8). All potentials reported in this work are referenced to the reversible hydrogen electrode, or RHE, unless noted otherwise. At 1 atm of CO, only a weak band attributable to linearly adsorbed CO ($CO_L$) was observed when the electrode potential was stepped from 0 to −0.9 V, which could be attributed to the rapid consumption of $CO_L$ by the CORR at this potential (Fig. 1a, b), in line with previous studies[35,36]. The size of the $CO_L$ peak remained largely unchanged when the CO pressure was increased from 1 atm up to 40 barg at −0.9 V (Fig. 1a). Intriguingly, if CO pressure was elevated at 0 V followed by stepping the electrode potential to −0.9 V, the $CO_L$ peak intensity exhibited a clear positive correlation with the CO pressure (Fig. 1b). All spectra were collected

after the intensity of the $CO_L$ peak stabilized for at least 15 min at any predetermined condition, i.e., CO pressure and electrode potential. These spectroscopic results were independently reproduced for at least three times. A lower partial current density (by 68.6%) and a lower Faradaic efficiency (FE) (by 31.6%) for the CORR were determined with CO pressurized to 30 barg at −0.9 V than those with CO pressurized to 30 barg at 0 V before switching to −0.9 V (Fig. 1c). Our recent investigations show that the $CO_L$ peak intensity is linearly correlated to its coverage on Cu within the pressure and potential windows investigated in this work[36]. Thus, the difference in reactivities could be attributable to the lower CO coverage when the CO pressure was elevated after the application of the negative potential. Similar impacts of the sequence of applying potential and elevating CO pressure on the CORR reactivities were observed with a higher overpotential at −1.0 V (Supplementary Fig. 1). No noticeable structural changes in post-reaction Cu electrode were observed (Supplementary Fig. 2), indicating the differences in the CO coverage and CORR reactivities are not attributable to the changes in the physical structure of Cu electrodes during reaction.

### Two-state model of the electric double layer

The theory of thermodynamics dictates that the same final state must be established − if the system is allowed to reach equilibrium − regardless of the paths taken between the same initial and final states. Intriguingly, the sequence in which the final state of the Cu surface at −0.9 V and 40 barg of CO is reached plays a key role in the $CO_L$ coverage of the final state. The stark difference in the $CO_L$ coverage is a strong evidence that at least one of the observed final states is not fully equilibrated. In other words, the sequence in applying the negative potential and elevating the CO pressure can create at least one kinetically trapped state that is stable on the experimental time scale of >120 min in elevating CO pressure from 1 atm to 40 barg (Supplementary Fig. 3). The increased chemical potential in the gas phase CO (by elevating its partial pressure) impacts the $CO_L$ coverage via two equilibria, i.e., dissolution and adsorption. The dissolution equilibrium between the gas phase and the dissolved CO in the bulk electrolyte is relatively facile, and is independent of the applied potential. Meanwhile, the adsorption equilibrium occurs in the EDL, and its structure could depend on the condition at which it is established. We hypo-

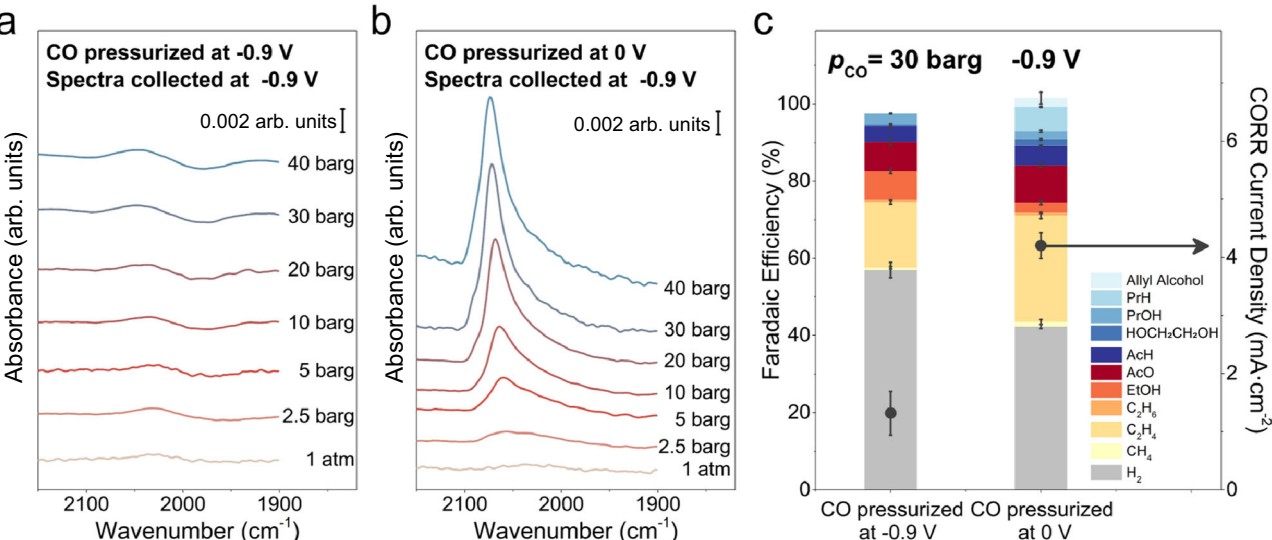

**Fig. 1 | Distinct pressure dependent $CO_L$ band and CO reduction reactivities when CO is pressurized at different potentials.** Pressure dependent $CO_L$ band on polycrystalline Cu at −0.9 V when the CO pressure is elevated at (**a**) −0.9 V and (**b**) 0 V in 0.1 M potassium phosphate buffer electrolyte (pH = 8). **c** Comparison of the CORR reactivities at $p_{CO}$ of 30 barg on polycrystalline Cu foil when the CO pressure is elevated at −0.9 V and 0 V in 0.1 M potassium phosphate buffer electrolyte. The error bars represent the standard deviation from at least three independent measurements.

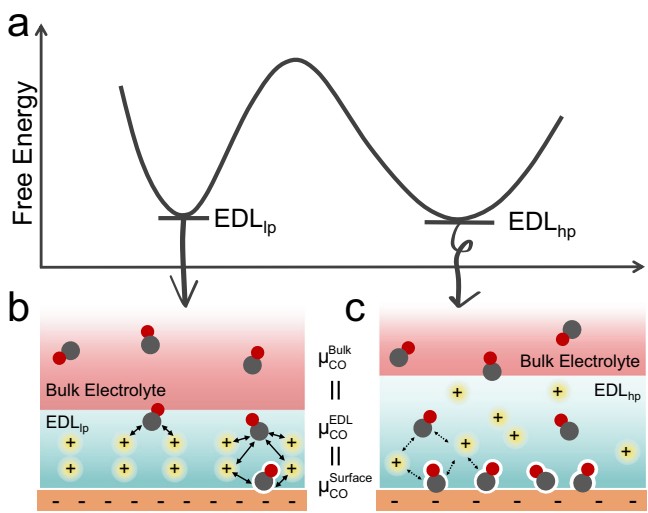

**Fig. 2 | Schematic representation of bistable states resulting from different sequences of applying negative electrode potential and elevating CO pressure.** **a** The EDL$_{lp}$ and the EDL$_{hp}$ are represented as individual local minima in the free energy landscape. **b** In the EDL$_{lp}$, the compact cation layer imposes strong repulsive interaction to incoming CO molecules, leading to lower CO concentration in the EDL$_{lp}$ and CO coverage on the surface. **c** In the EDL$_{hp}$, the repulsion is likely weaker due to the less compact EDL structure, leading to higher CO concentration and CO coverage.

thesize that the EDL established at −0.9 V with a lower CO pressure (i.e., EDL$_{lp}$, Fig. 2b) is more compact and rigid than that at a higher CO pressure (i.e., EDL$_{hp}$, Fig. 2c). Upon switching the potential from 0 to −0.9 V, a new state of EDL is established, in which cations are preferentially drawn to the interface due to the electrostatic attraction. Due to the presence of the strong interfacial electric field (−10$^7$ V/cm)[37,38], the composition and structure of EDL are distinct from the bulk electrolyte, and hence could be viewed as a separate phase from the bulk electrolyte. Our results demonstrate that the structure of EDL can be influenced by the CO concentration in the electrolyte, which is in equilibrium with the gas phase CO prior to the change of potential. In the EDL$_{lp}$, the cations and interfacial water molecules orient and interact largely in response to the interfacial electric field due to the low CO concentration when established. In contrast, the cations and interfacial water molecules must accommodate the presence of a higher density of dissolved CO in the EDL$_{hp}$, leading to a less compact and rigid structure. The lack of change in the CO$_L$ band with the EDL$_{lp}$ after the CO pressure is elevated suggests that the composition and structure of EDL$_{lp}$ are distinct from those of EDL$_{hp}$ even though the CO pressure and electrode potential are identical, and thus they can be considered as two different states (or phases) with distinct structures and properties (Fig. 2a). The presence of two distinct states at the same pressure and potential could be attributed to the insufficient thermodynamic driving force provided by the rise in the CO chemical potential (higher partial pressure) to overcome the activation barrier associated with the phase change from EDL$_{lp}$ to EDL$_{hp}$. In both EDL$_{lp}$ and EDL$_{hp}$, CO is likely in equilibrium with CO dissolved in bulk electrolyte due to the constant transfer of CO from the bulk electrolyte through the EDL to the Cu surface, as evidenced by the sustained CORR rate (Fig. 1c). The equilibria among CO in the gas phase, the bulk electrolyte, the EDL, and adsorbed on the Cu surface make their chemical potentials identical (Fig. 2b). Given that the chemical potentials of gas phase CO are identical in the scenarios forming the EDL$_{lp}$ and the EDL$_{hp}$ (i.e., identical final CO pressures), the chemical potential of adsorbed CO in the EDL$_{lp}$ and the EDL$_{hp}$ must also be equal ($\mu_{CO}^{EDL_{lp}} = \mu_{CO}^{EDL_{hp}}$) despite of drastically different coverages

($\theta_{CO}^{EDL_{lp}} \ll \theta_{CO}^{EDL_{hp}}$). It could be that a unit increase in $\theta_{CO}^{EDL}$ incurs a higher rise in the chemical potential of CO in the EDL$_{lp}$ than that of the EDL$_{hp}$ due to the higher energetic cost of disrupting the more rigid EDL$_{lp}$ structure. Another equivalent line of rationalization is that the repulsive interaction between dissolved/adsorbed CO and the more compact and rigid EDL$_{lp}$ is stronger than that in the EDL$_{hp}$.

### Rationalization of spectroscopic and reactivity results with the two-state model

The suppression of CO adsorption at the EDL$_{lp}$ is further demonstrated in potential dependent SEIRAS experiments. The kinetic barrier in the conversion between the EDL$_{lp}$ and the EDL$_{hp}$ is dependent on the potential at which the EDL is established. The further away from the potential of zero charge (PZC), the higher the kinetic barrier is expected to be, because the activation barrier associated with the phase transition in the presence of a stronger interfacial electric field is likely higher. We first prepare the EDL$_{lp}$ by pressurizing CO from 1 atm to 30 barg at −0.9 V, which is followed by anodic stepping of the electrode potential to 0 V before shifting the potential cathodically back to −0.9 V while maintaining the CO pressure at 30 barg (Fig. 3a). As the potential is stepped anodically, the CO$_L$ peak area grows up to −0.4 V, which could be attributed to the weakening interfacial electric field as the potential approaches the PZC. As the electrostatic attraction between the electrode and cations/interfacial water dipoles diminishes, it is energetically less costly to allow access of additional CO molecules into the EDL$_{lp}$, leading to the increase of CO$_L$ coverage on the Cu surface (Fig. 3b). When the potential is further increased to 0 V, the CO$_L$ peak area diminishes due to the weakened CO binding strength at less negative potentials and possible oxidation of Cu surface[39–42]. As the potential is subsequently shifted cathodically from 0 V, the CO$_L$ peak area increases until −0.4 V due to enhanced CO binding and the removal of possible oxygen-containing species on the Cu surface (Fig. 3a). Upon further decrease of electrode potential, the CO$_L$ peak area decreases slightly, likely because of the accelerated consumption by the CORR at larger overpotentials. Notably, the CO$_L$ peak area of the anodic scan is significantly smaller than that of the cathodic scan at the same electrode potential (Fig. 3b). This difference could be rationalized as follows. During the anodic potential steps, the EDL at each potential could be viewed as EDL$_{lp}$ at that potential because the CO partial pressure is 1 atm when the EDL is initially established. In contrast, during the cathodic potential steps, the EDL$_{hp}$ is established at each potential because the bulk electrolyte is presaturated with 30 barg of CO. Thus, the difference in the blue and red symbols at each potential represents the difference between the CO$_L$ coverage at the EDL$_{lp}$ and the EDL$_{hp}$. The ratio between the CO$_L$ peak size at the EDL$_{lp}$ and the EDL$_{hp}$ decreases from unity at 0 V to 0.22 at −0.9 V, which is consistent with the higher activation barrier for the phase transition at a more negative potential. Another likely cause contributing to the lack of the CO$_L$ band at −0.9 V in the EDL$_{lp}$ is the continuous consumption of CO$_L$ due to the CORR. Rate of the CORR at the EDL$_{hp}$ is higher than that of the EDL$_{lp}$ at −0.9 V, yet the CO$_L$ band is still visible at the EDL$_{hp}$. This strongly suggests that not only is the CO concentration in the EDL$_{lp}$ lower than that in the EDL$_{hp}$, but the transport of CO from the bulk electrolyte to the electrode surface is also more sluggish in the EDL$_{lp}$. If the CO transport through both the EDL$_{lp}$ and the EDL$_{hp}$ were identical, then the CO$_L$ coverage at the EDL$_{hp}$ would decrease over time due to its higher consumption rate. It is reasonable that CO is transported more efficiently through the less rigid and compact EDL$_{hp}$ compared to EDL$_{lp}$.

The dipole-coupling corrected Stark tuning rate is notably lower in the cathodic scan (43.2 cm$^{-1}$V$^{-1}$) compared to the anodic scan (61.1 cm$^{-1}$V$^{-1}$) (Fig. 3c, the dipole-coupling correction of CO$_L$ band wavenumbers details in Supplementary Note and Supplementary Fig. 4)[43]. This observation suggests a weaker electric field within the

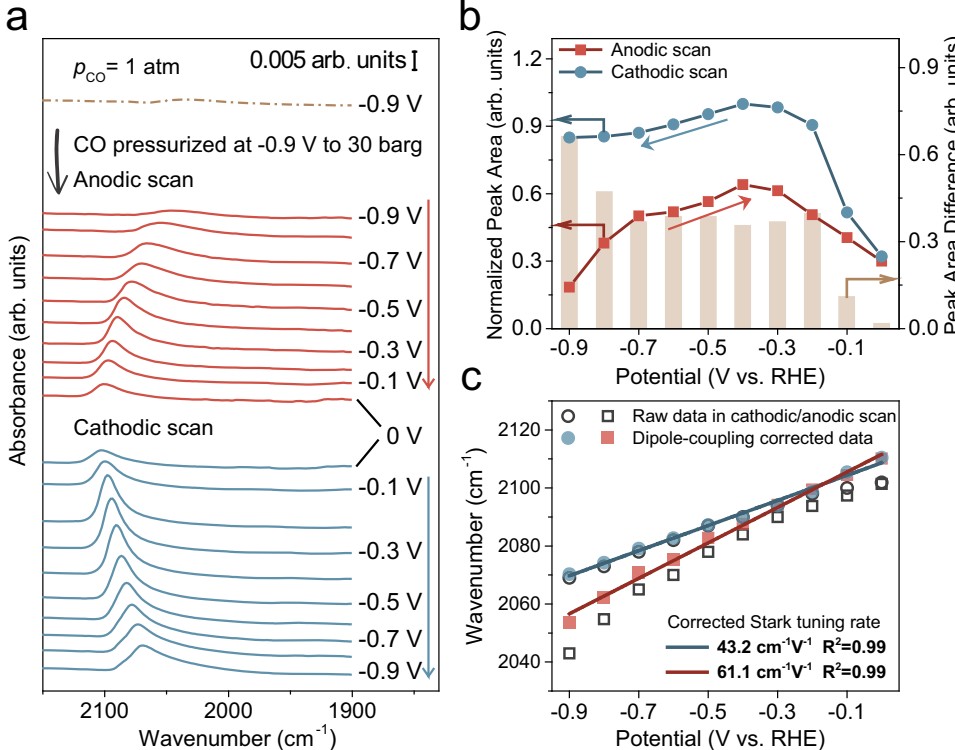

**Fig. 3 | Potential dependent $CO_L$ band and Stark tuning rates in anodic and cathodic scan. a** Potential dependent $CO_L$ band on polycrystalline Cu when potential stepping anodically after CO is pressurized to 30 barg at −0.9 V (red lines) and then stepping cathodically (blue lines) in 0.1 M phosphate buffer electrolyte of pH = 8. **b** Potential dependent $CO_L$ peak area in the anodic and the subsequent cathodic scan after CO is pressurized to 30 barg at −0.9 V. The maximum peak area was scaled to 1.0. The column represents the discrepancy in peak area at each potential. **c** Comparison of the dipole-coupling corrected Stark tuning rates of CO stretching frequency in anodic and cathodic scan.

$EDL_{hp}$ than the $EDL_{lp}$ at identical electrode potentials[44]. These results indicate a larger distance between the cations (the outer Helmholtz plane, OHP) and the electrode surface ($d_{OHP}$) within the $EDL_{hp}$ than that within the $EDL_{lp}$ by approximating the EDL as a planar capacitor[45]. This finding aligns with capacitance measurement at −0.3 V (a potential with no detectable CORR activity) that the $EDL_{hp}$ exhibits a much smaller specific capacitance compared to that of the $EDL_{lp}$ (Supplementary Fig. 5) as capacitance is inversely proportional to $d_{OHP}$[46]. Further, the smaller specific capacitance of the $EDL_{hp}$ compared to the $EDL_{lp}$ at identical electrode potential indicates a lower charge density, leading to the less compact cation layer in $EDL_{hp}$. This conclusion is consistent with the discussion above.

**Impact of different cations on the properties of EDL**
Cations play a decisive role in determining the rigidity of the EDL at negative potentials. Given the concentration of a dilute solute like CO could affect the structure of the EDL, it stands to reason for the identity of cations electrostatically drawn to the negatively charged electrode surface to be a key factor in determining the EDL properties. In this section, we investigate the impact of cation identity on the EDL structure. 18-Crown-6 is known to effectively chelate equimolar $K^+$ ions to form a substantially bulkier chelated complex, which is referred to as C-$K^+$ below[35,47]. Thus, introduction of 18-Crown-6 to a $K^+$ containing electrolyte could substantially alter the properties of cations. In the electrolyte with one-quarter of $K^+$ ions chelated by 18-crown-6 (crown ether: $K^+$ = 1:4), a clear, albeit weak, positive correlation between $CO_L$ peak intensity and $p_{CO}$ emerged as electrode potential is maintained at −0.9 V when the CO pressure is elevated, i.e., the $EDL_{lp}$ (Fig. 4a). This is a clear indication that replacing a quarter of $K^+$ with bulkier and more hydrophobic C-$K^+$ can make the $EDL_{lp}$-to-$EDL_{hp}$ transition feasible with the increase of $p_{CO}$ or reduce the structural difference between the

$EDL_{lp}$ and the $EDL_{hp}$ at −0.9 V. As expected, the $CO_L$ peak intensity exhibits a significantly stronger correlation to $p_{CO}$ when most of $K^+$ is chelated by the crown ether (crown ether: $K^+$ = 1:1) (Fig. 4b). The increased cation size and reduced interaction between $K^+$ and water due to the presence of the chelating crown ether likely make the EDL less rigid, thus making the EDL structure more responsive to $p_{CO}$. A similar trend was observed with tetramethylammonium (TMA$^+$), which is known to be larger and more hydrophobic than $K^+$ (Fig. 4c)[48,49].

The ratio between the integrated areas of the $CO_L$ peak collected with CO pressure raised to 40 barg with the $EDL_{lp}$ and the $EDL_{hp}$ ($R_{lp/hp}$) at −0.9 V offers a more quantitative measure of the rigidity of the EDL structure with different cations. A $R_{lp/hp}$ value of unity entails that the EDL structure is flexible and porous, perfectly responsive to $p_{CO}$ changes, e.g., the kinetic barrier of EDL phase change can be overcome at room temperature (~2.5 kJ/mol)[50,51]. In contrast, a $R_{lp/hp}$ value of close to zero shows that the EDL is compact and phase change cannot be induced by raising $p_{CO}$, which indicates a significant kinetic barrier to change the state of EDL at room temperature. This corresponds to an EDL with a rigid structure that imposes strong repulsive interactions to the incoming CO molecules from the bulk electrolyte. We determined the $R_{lp/hp}$ ratio in seven different cations with varying sizes (Fig. 4d, e, and Supplementary Fig. 6). The $R_{lp/hp}$ ratio increases almost linearly with the radii of hydrated cations (Fig. 4e)[47,49], suggesting that the size of the hydrated cation plays a decisive role in determining the rigidity of the EDL. EDLs containing smaller hydrated cations such as Cs$^+$ and K$^+$ are more rigid than those with bulkier ones. One likely reason for this correlation is that smaller hydrated cations allow for tighter packing, leading to a stronger interfacial electric field and making structure changes energetically costlier. In addition, smaller hydrated cations are generally considered as structure-breaking, while the larger ones are likelier to be structure-making[52,53]. Structure-making cations either

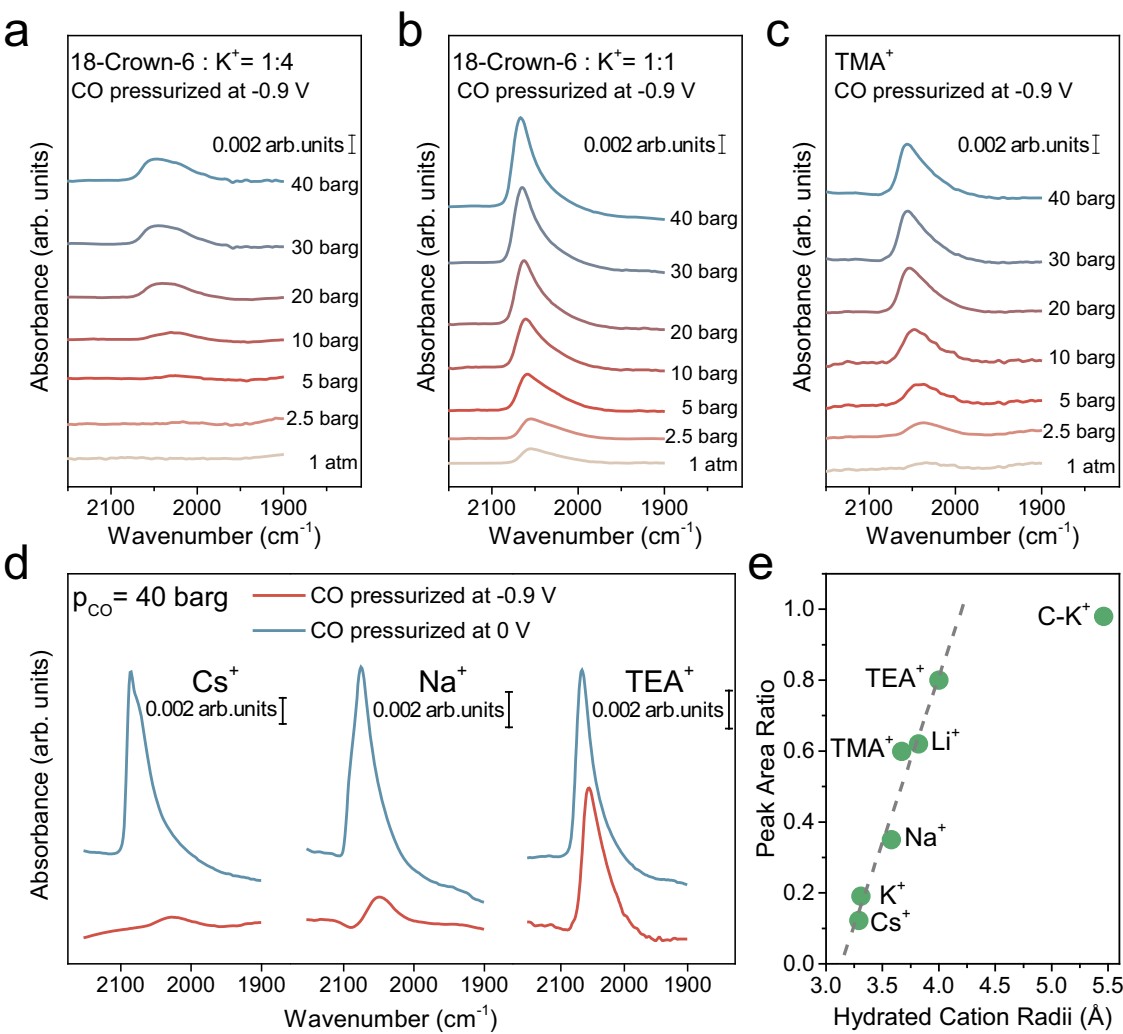

**Fig. 4 | Suppression of CO adsorption with cations of different size.** Pressure dependent $CO_L$ band when CO is pressurized at −0.9 V for molar ratio of 18-crown-6 and K+ is (**a**) 1:4, (**b**) 1:1 and for (**c**) TMA+. **d** Comparison of ATR-SEIRAS spectra of $CO_L$ band at 40 barg in electrolytes containing various cations, with CO pressurized at −0.9 V (red) and 0 V (blue). **e** Plots the peak area ratio of $CO_L$ band when CO is pressurized to 40 barg at −0.9 V and 0 V as a function of the cation's hydrated radius for seven cations used in this study.

have strongly bound hydration shells, e.g., Li+, or very weakly bound hydration shells, if at all, e.g., tetraalkylammonium cations. A common feature of structure-making cations is that they interact relatively weakly with water molecules outside of their hydration shells, which tends to reduce connectivity between the hydrated cations and the rest of the water molecules[52,54]. In contrast, in addition to water molecules in the hydration shell, structure-breaking cations tend to maintain relatively strong interactions with surrounding water molecules, but are not strong enough to significantly disrupt the hydrogen bonds between water molecules in the hydration shells and the next layer of water. This configuration makes hydrated structure-breaking cations, e.g., Cs+ and K+, well connected with surrounding water, thus leading to more structurally rigid EDLs.

Larger hydrated cations tend to exhibit lower CORR activity at elevated CO pressure (Supplementary Fig. 7), even though they can provide a higher CO coverage by forming less compact and rigid EDL. This is consistent with our previous study at ambient pressure[35]. The weaker interfacial electric field associated with the loosely packed bulkier cations at a given electrode potential could be a key factor in affecting the CORR activity. Our recent work suggests that the strength of the interfacial electric field alone cannot account for the cation effect in its entirety. Short-range chemical interactions between the surface-bound reaction intermediate and cations were suggested as

another potential cause[55]. In light of the varied rigidity of the EDL with different cations, the flexibility of interfacial water − the dominant proton donor in the CORR − in the EDL could be a significant contributing factor.

## Discussion
In this work, we provide experimental evidence that the EDL could be viewed as a separate phase from and is not always in the equilibrium state. In situ spectroscopic results of CO adsorption on Cu surfaces show that the sequence at which negative potential and elevated CO pressure are applied has a clear impact on the coverage of CO and the reactivity of the CORR. The structure of EDL established at a low CO partial pressure and negative potential could be kinetically trapped even when the CO partial pressure is elevated. A two-state model, in which EDLs established at the same negative potential but different CO pressures represent two different phases, is proposed to rationalize the spectroscopic observations. Further, we show that the identity, especially the size of the hydrated cations has a decisive impact on the rigidity of the EDL phase, with the EDL containing smaller hydrated cations being more rigid and less susceptible to phase change. The rigidity of the EDL is likely to be a key factor in affecting the electro-catalytic performance, and thus results reported in this work introduce a new dimension in understanding the EDL structure and the cation

effect. This study conducted in H-cell configuration provides a foundational model elucidating the impacts of the double layer structure on CO adsorption and electrocatalysis. Further investigations of these phenomena in reactor configurations close to practical devices, e.g., the flow configuration using gas diffusion electrode, could be a fruitful research direction in gaining the mechanistic understanding of double layer effect at the triple-phase interface. It is crucial to recognize that the rigidity of EDL structure is intimately associated with the electrostatic forces within it, which are notably affected by the pH of the electrolyte[56,57]. This specific interaction between pH and EDL structure presents an intriguing area for future research.

## Methods

### Materials
Cu foil (0.1 mm thick, 99.9999% metal basis), lithium hydroxide monohydrate (99.995% metals basis), tetraethylammonium hydroxide (35 wt% aqueous solution) and dimethyl sulfoxide (≥99.9%) were purchased from Alfa Aesar. Potassium hydroxide (semiconductor grade, 99.99% trace metals basis), sodium hydroxide (semiconductor grade, 99.99% trace metals basis), cesium hydroxide monohydrate (99.95% trace metals basis), phosphoric acid (crystalline, 99.999%, trace metals basis), phosphoric acid (ACS reagent, ≥85 wt% in $H_2O$), Chelex 100 sodium form, deuterium oxide (99.9 atom% D), potassium chloride (99.5%), ammonium fluoride solution (40%), hydrogen fluoride (99%), sodium tetrachloroaurate dihydrate (99%), ammonium chloride (99.99%), sodium thiosulfate pentahydrate (98%), cupric sulfate (99.99% trace metal basis), sulfuric acid (99%) graphite rod (99.999%) were purchased from Sigma Aldrich. 18-Crown-6 (99.0%), tetramethylammonium hydroxide pentahydrate (98%) were purchased from Acros Organics. Sodium carbonate (99.999%) was purchased from Merck. 4 M KCl saturated with AgCl was purchased from Fisher Scientific. Nafion 117 membrane, Nafion solution (10 wt%) were purchased from Fuel Cell Store. Carbon monoxide (99.999%) and argon (99.999%) were purchased from Air Liquide. All electrolyte solutions were prepared with Milli-Q water (18.2 MΩ cm) and were treated with Chelex to remove trace metal residues prior to electrolysis.

### High-pressure electrochemical cells
High-pressure CO electroreduction investigations were carried out in custom-designed two-compartment, three-electrode electrochemical cells capable of operating at pressures up to 60 barg. The cathode and anode chambers were separated by a Nafion 117 membrane to prevent product crossover between compartments. The working, reference, and counter electrodes were connected to the lids with embedded electrical contact. A gas dispersion tube was mounted on the cathode chamber lid to connect with the inlet gas pipeline and to deliver CO into the electrolyte. The air tightness of the electrochemical cells was ensured by EPDM rubber O-rings and machine screws with proper torque. The schematic and the image of the high-pressure surface-enhanced infrared absorption spectroelectrochemical cell are depicted in Supplementary Fig. 8. Detailed information on the reactors' design and operational procedures can be found in our previous publication[36].

### Electrode preparation
For spectroscopic experiments, the Au film on the silicon ATR crystal was first prepared with established protocols in previous works[58,59]. The ink consisting polycrystalline Cu powder was prepared by dispersing 100 mg commercial Cu powder in 2.5 ml isopropanol with 30 μL of Nafion solution. After sonicating for 30 min, 50 μL of the ink was uniformly dropped onto 1 cm² of the Au film, followed by drying at 55 °C for 1 h, which was used as the working electrode in SEIRAS experiments[34,43]. The mass loading of Cu powder electrode is 2 mg cm⁻². To exclude the potential influence of Nafion ionomer to the CO adsorption behavior, electrodeposited Cu films on Au-coated silicon ATR crystals were also prepared. Cu film electrodeposition was

conducted at a constant current of −260 μA cm⁻² for 8 min in an Ar-saturated solution containing 5 mM $CuSO_4$ and 50 mM $H_2SO_4$ following previous works[34,60]. Both Cu power and Cu film electrodes exhibited consistent CO adsorption behavior at elevated CO pressures (Fig. 1 and Supplementary Fig. 9), suggesting that Nafion has a minimal impact on CO adsorption under our specified electrochemical conditions. For reactivity experiments, the polycrystalline copper foil working electrode was polished using sandpaper (P1200, STARCKE), followed by electropolishing in 85% ortho-phosphoric acid at 2.1 V versus a graphite rod counter electrode for 5 min, and thoroughly rinsed with Milli-Q water. The counter electrode was an iridium-coated titanium foam and the fabrication procedures were described in a published literature[61].

### Electrolyte preparation
The phosphate buffer of pH 8 was prepared by dissolving 0.19 M MOH ($M^+$ = $K^+$, $Na^+$, $Cs^+$, $TMA^+$, $TEA^+$) and 0.1 M $H_3PO_4$ in Milli-Q water and titrated with MOH or $H_3PO_4$ to achieve pH of 8 measured with an Orion Star™ A111 Benchtop pH Meter (Thermo Fisher Scientific). For electrolyte containing $Li^+$, lithium carbonate buffer solution was used due to the low solubility of lithium phosphate. Lithium carbonate buffer was prepared by dissolving 0.19 M LiOH in Milli-Q water and purging with $CO_2$ to achieve pH of 8. For the experiments with crown ether, 18-Crown-6 was directly added to the potassium phosphate electrolyte with a pre-determined molar ratio of 18-Crown-6 to $K^+$.

### High-pressure CO electroreduction experiments
Before each experiment, Teflon and PEEK compartments were cleaned with aqua regia and thoroughly rinsed with Milli-Q water. Before electrolysis, the electrolyte was purged with CO for 5 min. Subsequently, the outlet valves were closed to achieve the desired pressure by adjusting the pressure regulator. Upon reaching the specified pressure, the inlet valves were closed, allowing the system to stabilize for 15 min to ensure gas-liquid equilibrium. Mechanical stirring was maintained at a rate of 1500 rpm.

For spectroscopic experiments, the electrode potential was controlled by a Bio-Logic SP-150 potentiostat and the solution resistance was measured with the Current Interrupt (CI) method. During the SEIRAS experiment, the potentiostat compensated 85% of the resistance, with the remaining 15% corrected manually to arrive at the actual potentials.

For reactivity assessments, a Gamry Reference 600+ Potentiostat was employed. Potentiostatic electrochemical impedance spectroscopy (PEIS) was executed from $10^5$ to $10^3$ Hz at electrolysis potential to measure the solution resistance. The potentiostat was set to compensate 85% of the value and the remaining was manually corrected afterward to arrive at the actual potentials.

The measured potential was converted to the RHE reference scale using the formulas E (vs. RHE) = E (vs. Ag/ AgCl) + 0.197 V + 0.0591 V × pH. Every reactivity data point was the average of at least three independent electrolysis experiments, based on which the standard deviations (SD) were calculated.

CO electroreduction experiments were conducted at two distinct pressurization conditions: 1) CO was pressurized at the intended electrode potential where CORR occurred, such as -0.9 V or −1.0 V. 2) CO was pressurized at 0 V, and then CORR reactivity was investigated after switching the electrode potential to −0.9 V or -1.0 V.

### SEIRAS experiment
In a typical SEIRAS experiment, the spectroelectrochemical cell was placed in the sample compartment with custom-built light pathway of a Bruker Invenio spectrometer equipped with a liquid nitrogen-cooled MCT detector. The pressure dependence of $CO_L$ was investigated at different pressurization conditions. 1) The electrode potential was initially set to −0.9 V at 1 atm CO. CO was then pressurized to a fixed

$p_{CO}$ while maintaining the electrode potential of −0.9 V. The system was kept for 15 min under each pressure before spectra were collected. 2) CO was pressurized to a fixed $p_{CO}$ at 0 V and equilibrate for 15 min. Afterward, the electrode potential was shifted to −0.9 V and spectra were then collected.

## Products detection and quantification

After each CO electroreduction experiment, the headspace gas from the two compartments was released simultaneously by adjusting the needle valves carefully. The sample gas of the cathode chamber was collected in a gas sampling bag with appropriate size depending on the CO pressure and further introduced into the sample loop of a gas chromatograph (Agilent 7890B). This gas chromatograph was equipped with a Shin-Carbon ST Micropacked GC column and a Hayesep Q column. Argon served as the carrier gas. Using a thermal conductive detector (TCD), $H_2$ was analyzed. CO, $CH_4$, $C_2H_4$, and $C_2H_6$ were assessed using a flame ionization detector (FID) equipped with a methanizer. To determine the quantity of gaseous products, the CO pressure, with a compressibility factor of 1, and a headspace volume of 52 ml were considered. Liquid products were analyzed using a Bruker AVIII 600 MHz NMR spectrometer. After electrolysis, a mixture of 500 μL electrolyte and 100 μL of $D_2O$ was prepared. An internal standard of 50 μL dimethyl sulfoxide (≥99.9%, Alfa Aesar) was added. The 1H spectrum was measured with water suppression by using an excitation sculpting method.

## Double layer capacitance measurement

The electric double layer capacitances of $EDL_{lp}$ and $EDL_{hp}$ were measured by potential step voltammetry analyzing transient current response to a voltage pulse and electrochemical impedance spectroscopy (EIS) at −0.3 V without CORR[62,63]. The electrochemical cell was first pressurized with CO at either −0.3 V or 0 V and was kept for 15 min to reach the dissolution and adsorption equilibria. The mechanical stirring rate was set at 1500 rpm. Subsequently, the electrode potential of −0.3 V was kept or applied for 10 s, followed by stepping to a more cathodic potential of −0.31 V for 0.02 s. Exponential discharge current density was observed due to the voltage pulse (ΔE) of 10 mV (Supplementary Fig. 5). The released charge per surface area (Δq) was quantified by integrating the transient current density vs. t plot. The specific double layer capacitance ($C_{dl}$) was then calculated according to the following equation:

$$C_{dl} = \frac{\Delta q}{\Delta E} \tag{1}$$

In EIS measurement, the electrochemical cell was first pressurized with CO at either −0.3 V or 0 V to 30 barg and allowed to stabilize for 15 min to reach the dissolution and adsorption equilibria. Subsequently, the EIS measurements were conducted at −0.3 V using a Gamry Reference 600+ Potentiostat within the frequency range between 30 kHz and 1 Hz and 10 mV amplitude. The obtained Nyquist plots were fitted by Zview software with an equivalent circuit of $R_s$–$R_{ct}$/CPE (Supplementary Fig. 5, Supplementary Table 1), where $R_s$ is the solution resistance and $R_{ct}$ is the charge transfer resistance. The capacitive effects of EDL were modeled using a constant phase element (CPE) to offset the influence of the distributed time constant. The impedance of CPE is $1/Y_0(j\omega)^n$, where $Y_0$ is the CPE coefficient and the exponent, n, is in the range of $0 \leq n \leq 1$[15,63]. The equivalent capacitance was calculated using the method reported by Brug et al. with Eq. (2)[64].

$$C_{eff} = Y_0^{\frac{1}{n}} \cdot \left( \frac{1}{R_u} + \frac{1}{R_{ct}} \right)^{\frac{n-1}{n}} \tag{2}$$

The specific $C_{EDL}$ was then determined by normalizing $C_{eff}$ to the geometric surface area of Cu foil electrode, which was determined to be 2.2 cm².

## Data availability
The data that support the findings of this study are available from the corresponding author upon request. Source data are provided with this paper.

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

## Acknowledgements

J.H. and Q.L. acknowledge the financial support from the State Key Laboratory of Chemical Engineering (No. SKL-ChE-23T02). B.X. acknowledges the financial support from Beijing National Laboratory for Molecular Sciences. All NMR experiments were performed at the BioNMR facility, Tsinghua University Branch of China National Center for Protein Sciences (Beijing). The authors thank Dr. Ning Xu for assistance in NMR data collection.

## Author contributions

J.H., B.X., and Q.L. conceived and designed this project. J.H. carried out the experiments. All authors contributed to data analysis and writing of this manuscript.

## Competing interests

The authors declare no competing interests.

## Additional information

# QUERY FORM

| NATURECOMMUNICATIONS | |
|---|---|
| **Manuscript ID** | **[Art. Id: 46318]** |
| **Author** | |
| **Editor** | |
| **Publisher** | |

## Journal: NATURECOMMUNICATIONS

**Author** :- The following queries have arisen during the editing of your manuscript. Please answer by making the requisite corrections directly in the e.proofing tool rather than marking them up on the PDF. This will ensure that your corrections are incorporated accurately and that your paper is published as quickly as possible.

| Query No. | Description | Author's Response |
|---|---|---|
| | No queries | |

