## [Peer Review File · Nature Communications]

Influence of Electric Double Layer Rigidity on CO Adsorption and Electroreduction RateREVIEWER COMMENTS

Reviewer #1 (Remarks to the Author):

The authors have combined CORR reactivity measurements and the high-pressure in-situ surface enhanced spectroscopy to demonstrate that the adsorption of CO_{ad} can be profoundly influenced by the structure of EDL, which is dependent on the order of applying negative electrode potential and elevating CO pressure. This work firstly provides the experimental evidence for a non-equilibrated EDL structure and its impact on electrocatalytic performance. I would recommend this work for publication in Nature Communications, and the following are some comments that should be considered before publication:

1. The CO coverage on Cu surfaces and the reactivity of the CORR are attributed to the structure of EDL established at different conditions in this article. However, the authors should consider whether the sequence at which negative potential and elevated CO pressure are applied might affect the structure of Cu. This could potentially lead to differences in CO adsorption.
2. The commercial Cu powder was used as the working electrode in SEIRAS experiments, while, for reactivity experiments, the polycrystalline copper foil working electrode was used. I am concerned whether the different electrodes will affect the reliability of the results.
3. The system tested is in an H-cell, and I don't understand the significance of this model in practical applications. Currently, CO(2)RR has been advanced to use a membrane electrode system, can this phenomenon still exist in the three-phase interface of a gas diffusion electrode?
4. Of the two bilayer models given in the paper, it is ultimately the CO mass transfer that triggers it. I am concerned about why EDL_{ip} is stabilized, and it should be explored more deeply and detailed evidence should be given, not just some speculations. Because when EDL_{ip} dissolves enough CO and is driven by an applied voltage, it is hard to be convinced that the chemical potential alone maintains this unstable state.
5. Can the authors provide the details and schematics of devices for high-pressure in-situ surface enhanced spectroscopy?
6. As we all know EDL is an important but complicated problem in electrochemical reactions. Some theoretical explorations or discussions in this work are necessary to quantitatively understand the origin of electric double-layer rigidity affecting on CO adsorption and some key reaction steps. Some excellent theoretical works can be referenced, e.g., Sci. Adv. 2020. 6, eabb1219, J. Am. Chem. Soc. 2023, 145, 26, 14267.

Reviewer #2 (Remarks to the Author):

In this manuscript, Prof. Lu and co-workers performed rigorous in situ SEIRAS studies under different potentials and CO pressures. Their main argument is the presence of two kinetically stable EDL structures and their effects on CO coverage and subsequent CO reduction activity.

This reviewer enjoyed reading the manuscript and would like to congratulate the authors for their impressive work. However, please note the following comments, one of which may be critical and needs to be clarified before its publication.

A critical issue raised by this reviewer was found in the experimental section. The authors prepared an electrode with the Nafion ionomer to immobilize Cu powder on the Si ATR. Before realizing the use of the Nafion ionomer, this reviewer simply assumed an EDL structure relevant to the ideal GCS model, although the model was developed for electrolytes with low salt concentration. However, in the experiment, a thin layer of Nafion was found between the Cu and electrolyte interfaces, which is different from the assumed GCS model. The presence of the Nafion was also omitted in the schematic representation of Figure 2. In the literature, the importance of the electrode-ionomer interface in electrocatalysis has been suggested and its different EDL properties have also been presented. In the performance analysis, bare Cu without ionomer was seemingly used. Therefore, to clarify the authors' argument, it may be necessary to perform identical experiments with Cu electrode on ATR without

Nafion ionomer.

The following comments are my personal questions or some minor questions.

In situ SEIRAS analysis during potential swing from -0.9 to 0 and back to -0.9, and that with cation chelating agent or organic cation strongly support the authors' arguments. Another effective strategy to control the EDL structure will be to control the pH of the electrolyte, which changes the potential difference apart from the PZC of the Cu electrode. Could the authors show how the electrolyte pH affects the CO coverage or SEIRAS signals during the CO reduction reaction?

In the EDL_{lp} and EDL_{hp} conditions, different production rates of CO reduction are shown in Figure 1c. However, these differences seem to be limited to H₂ and ethylene production rates and not to other products. If CO coverage determines the overall production rates, why are those of other products almost unaffected?

The authors measured the double layer capacitance simply by dividing the released charge by the voltage pulse. However, this may not be a general method to estimate EDL capacitance, but EIS method has been commonly used. If the capacitance is estimated by the EIS method, will the authors get identical results?

Some typos (e.g. the 4th line of the second paragraph of the introduction) should be corrected.

Response to Reviewer #1 (Remarks to the Author):

The authors have combined CORR reactivity measurements and the high-pressure in-situ surface enhanced spectroscopy to demonstrate that the adsorption of CO_{ad} can be profoundly influenced by the structure of EDL, which is dependent on the order of applying negative electrode potential and elevating CO pressure. This work firstly provides the experimental evidence for a non-equilibrated EDL structure and its impact on electrocatalytic performance. I would recommend this work for publication in Nature Communications, and the following are some comments that should be considered before publication:

Response:

We thank the reviewer #1 for the positive assessment.

1. The CO coverage on Cu surfaces and the reactivity of the CORR are attributed to the structure of EDL established at different conditions in this article. However, the authors should consider whether the sequence at which negative potential and elevated CO pressure are applied might affect the structure of Cu. This could potentially lead to differences in CO adsorption.

Response:

We acknowledge Reviewer #1's concern regarding the potential impact of the sequence of applying negative potential and elevated CO pressure on the structure of Cu. In this revision, we collected SEM images of polycrystalline Cu powder electrode utilized in spectroscopic studies and AFM images of the polycrystalline Cu foil electrode utilized in reactivity studies with different sequences of applying potential and pressure. These sequences include (1) pressurizing CO to 40 barg at -0.9 V and (2) pressurizing CO to 40 barg at 0 V before applying an electrode potential of -0.9 V. Both SEM and AFM images of the post-experiment polycrystalline Cu electrodes show no noticeable structure difference regardless of the sequences (Figure R1), which suggests that the variations in CO coverage and reactivity observed in our study are not caused by changes in the physical structure of the Cu electrodes.

Figure R1. The field-emission scanning electron microscopy images of post-experimental (a-d) Cu powder recorded on a Merlin FESEM from Zeiss and AFM height images of post-experimental (e, f) Cu foil obtained using Bruker Dimension ICON where (a, c, e) CO was pressurized at -0.9 V and (b, d, f) CO was pressurized at 0 V.

Action:

We added the following sentence to the end of the paragraph on page 3: “No noticeable structural changes in post-reaction Cu electrode were observed (Supplementary Fig. S2), indicating the differences in the CO coverage and CORR reactivities are not attributable to the changes in the physical structure of Cu electrodes during reaction.”

We also added Figure R1 into the SI as Supplementary Fig. S2.

Supplementary Fig. 2. The field-emission scanning electron microscopy images of post-experimental (a-d) Cu powder recorded on a Merlin FESEM from Zeiss and AFM height images of post-experimental (e, f) Cu foil obtained using Bruker Dimension ICON where (a, c, e) CO was pressurized at -0.9 V and (b, d, f) CO was pressurized at 0 V.

2. The commercial Cu powder was used as the working electrode in SEIRAS experiments, while, for reactivity experiments, the polycrystalline copper foil working electrode was used. I am concerned whether the different electrodes will affect the

reliability of the results.

Response:

We understand the reviewer's concern regarding the use of different Cu electrodes in SEIRAS and reactivity investigations. To address this concern, we conducted reactivity measurements using the same Cu powder as in our SEIRAS experiments under two specific conditions: (1) pressurizing CO to 30 barg at -1.0 V and (2) pressurizing CO to 30 barg at 0 V before applying an electrode potential of -1.0 V. The working electrodes were prepared by airbrushing an ink containing Cu powder, isopropanol, and Nafion onto Sigracet 39 AA carbon fiber paper substrate. As shown in Figure R2, a significantly lower CORR rate and Faradaic efficiency were observed at the Cu powder electrode when CO was pressurized at -1.0 V. This finding aligns with the results obtained using the Cu foil electrode, affirming the consistency of our experimental outcomes. The observed decrease in CORR rate and Faradaic efficiency on Cu powders, in comparison to the Cu foil, regardless of the sequence of applying potential and pressure is likely due to the exposure of the carbon fiber paper support to the electrolyte, which enhances the competing hydrogen evolution reaction (HER).

Figure R2. Comparison of CO reduction reactivities at -1.0 V and p_{CO} of 30 barg on Cu powder electrode when CO is pressurized at -1.0 V and 0 V in 0.1 M potassium

phosphate buffer electrolyte of pH=8.

Action:

We added Figure R2 into the SI as the Supplementary Fig. 1.

Supplementary Fig. 1. Comparison of CO reduction reactivities at -1.0 V and p_{CO} of 30 barg on (a) polycrystalline Cu foil and (b) Cu powder electrodes when CO is pressurized at -1.0 V and 0 V in 0.1 M potassium phosphate buffer electrolyte of pH=8. The observed decrease in CORR rate and Faradaic efficiency, in comparison to the Cu foil, is likely due to the exposure of the carbon fiber paper support to the electrolyte, which enhances the competing hydrogen evolution reaction. The error bars represent the standard deviation from at least three independent measurements.

3. The system tested is in an H-cell, and I don't understand the significance of this model in practical applications. Currently, CO₂RR has been advanced to use a membrane electrode system, can this phenomenon still exist in the three-phase interface of a gas diffusion electrode?

Response:

With recent advancement, the rate of CO₂RR can be significantly enhanced in a membrane electrode system due to improved mass transport at the triple-phase-boundary site. Current understanding suggests that the key difference in electrode-electrolyte interface at triple-phase-boundary, compared to the H-cell configuration, is its effectively reduced electrolyte diffusion layer, which is approximately two orders of magnitude smaller (*ACS Energy Lett.* 2018, 3, 4, 855–860; *ACS Catal.* 2019, 9, 6, 4709–4718). Despite being more complex, it is unlikely that the double layer structure at the electrode-electrolyte interface with a reduced diffusion layer would deviates

drastically from the classic Gouy-Chapman-Stern model. In the mechanistic study of electrochemical interface, the electrode potential needs to be precisely monitored and controlled as it is the driving force in the establishment such interface. The employment of a reliable reference electrode remains challenging in membrane electrode systems. Moreover, development of in-situ spectroscopic techniques suitable for these systems is still at an early stage, hampering mechanistic analysis. Thus, insights gained in studies using H-cell remain informative for not only the conventional solid/liquid interface, but also more complex electrochemical interfaces with polymer electrolytes.

Action:

We added on page 10, at the end of the conclusion section: “This study conducted in H-cell configuration provides a foundational model elucidating the impacts of the double layer structure on CO adsorption and electrocatalysis. Further investigations of these phenomena in reactor configurations close to practical devices, e.g., the flow configuration using gas diffusion electrode, could be a fruitful research direction in gaining mechanistic understanding of double layer effect at triple-phase interface.”

4. Of the two bilayer models given in the paper, it is ultimately the CO mass transfer that triggers it. I am concerned about why EDL_{lp} is stabilized, and it should be explored more deeply and detailed evidence should be given, not just some speculations. Because when EDL_{lp} dissolves enough CO and is driven by an applied voltage, it is hard to be convinced that the chemical potential alone maintains this unstable state.

Response:

We appreciate the concern raised by this reviewer. The ability of EDL_{lp} to hold its structure even after the CO pressure was raised from 1 atm to 30 barg initially came as a surprise to us, and was a key piece of evidence leading to the proposed two-state model in this work. Once EDL_{lp} is established upon applying the negative potential, the intensity of the CO peak and the CORR rates remained unchanged over a duration of at least 2 hours. This was evident in Figure R3, where two EDL_{lp} were established by pressurizing CO at -0.6 V (Figure R3a) and -0.9 V (Figure R3b), respectively. The CO_L band area (S₁) remained stable throughout the 2-hour experimental period. Upon shifting the potential to 0 V and then back to -0.6 or -0.9 V, a marked increase in CO_L peak area (S₂) was observed, confirming that EDL_{lp} indeed suppressed the adsorption of CO. Further, the more negative the potential at which the EDL_{lp} is established, the more severely the CO_L band is suppressed, as shown by the much smaller CO_L band when the lower bound potential of -0.9 V compared to that of -0.6 V (Figure R3a,b). In reactivity measurement where EDL_{lp} was established by pressurizing CO at -1.0 V, both the total current density and the CORR Faradaic efficiency exhibited consistent

stability over 2 hours (Figure R3c), indicating a stable EDL_{lp} at this condition. A higher and stable CORR rate was observed when CO was pressurized at 0 V (Figure R3d), suggesting that the higher CO coverage in EDL_{hp} is beneficial to the CORR rate. Our findings suggest that EDL_{lp} is stable over the timescale of our experiments (> 2 hours), which was attributed to a significant barrier in the interconversion between EDL_{lp} and EDL_{hp} as depicted in Fig. 2a.

Figure R3. Time dependence of CO_L bands in SEIRAS investigations when CO is pressurized at (a) -0.6 V and (b) -0.9 V in 0.1 M potassium phosphate buffer electrolyte of pH=8. Total current density profiles and time dependent CORR Faradaic efficiencies for 2-hour electrolysis on Cu powder electrodes at -1.0 V and 40 barg when CO is pressurized at (c) -1.0 V and (d) 0 V in 0.1 M potassium phosphate buffer electrolyte of pH=8.

Action:

We modified the discussion on page 4 into: “In other words, the sequence in applying the negative potential and elevating the CO pressure can create at least one kinetically trapped state that is stable on the experimental time scale of > 120 min in elevating CO pressure from 1 atm to 40 barg (Supplementary Fig. 3)”.

Figure R3 is now added to Supplementary Information as Supplementary Fig. 3.

Supplementary Fig. 3. Time dependence of CO_L bands in SEIRAS investigations when CO is pressurized at (a) -0.6 V and (b) -0.9 V in 0.1 M potassium phosphate buffer electrolyte of pH=8. The CO_L band remained stable throughout at least 2-hour experimental period. Upon shifting the potential to 0 V and then back to -0.6 or -0.9 V,

a marked increase in CO_L peak area was observed. Total current density profiles and time dependent CORR Faradaic efficiencies for 2-hour electrolysis on Cu powder electrodes at -1.0 V and 40 barg when CO is pressurized at (c) -1.0 V and (d) 0 V in 0.1 M potassium phosphate buffer electrolyte of pH=8. These results suggest that EDL_{lp} is stable over the timescale of our experiments (> 2 hours), which is attributable to the significant barrier between EDL_{lp} and EDL_{hp} as depicted in Fig. 2a.

5. Can the authors provide the details and schematics of devices for high-pressure in-situ surface enhanced spectroscopy?

Response:

In this revision, we have provided a detailed description of our high-pressure in-situ surface-enhanced infrared absorption spectroscopy device, as depicted in Figure R4. The spectroelectrochemical cell is composed of two chambers separated by a Nafion 117 membrane sealed with two PTFE gaskets. Each chamber includes a lid and a Teflon cell body enclosed in titanium. Fittings to connect electrodes, inlet and outlet gas lines, a pressure gauge, a safety valve, and a CO gas dispersion tube are mounted on the lid. The gas tightness of the cell is ensured by rubber O-rings placed between the cell components. The lids and the Si crystal were assembled onto the cell bodies using screws, adjusted to a torque of 7 Nm. During SEIRAS experiments, CO gas was simultaneously delivered to the two compartments using two inlet lines connected with a T-junction. The outlet lines were connected to a T-junction and a needle valve to control the flow rate when releasing the high-pressure headspace gas after experiments.

Figure R4. (a) Schematic and (b) image of the custom-designed high-pressure surface enhanced infrared absorption spectroelectrochemical cell.

Action:

We added the following sentence in “High pressure electrochemical cells” section on page 11: “The schematic and the image of the high-pressure surface enhanced infrared absorption spectroelectrochemical cell is depicted in Supplementary Fig. 8.”

Figure R4 is added to the SI as Supplementary Fig. 8.

Supplementary Fig. 8. (a) Schematic and (b) image of the custom-designed high-pressure surface enhanced infrared absorption spectroelectrochemical cell.

6. As we all know EDL is an important but complicated problem in electrochemical reactions. Some theoretical explorations or discussions in this work are necessary to quantitatively understand the origin of electric double-layer rigidity affecting on CO adsorption and some key reaction steps. Some excellent theoretical works can be referenced, e.g., Sci. Adv. 2020. 6, eabb1219, J. Am. Chem. Soc. 2023, 145, 26, 14267.

Response:

We thank the reviewer for the suggestion. In response, we have cited the theoretical works mentioned by this reviewer to support the idea that applied potential can impact adsorbed reactive species within the EDL, thereby affecting the electrocatalytic reactions.

Action:

We have now added the following sentence and cited the above-mentioned references on page 2, first paragraph, line 6: “It has been established that the applied potential can

significantly influence adsorbed reactive species within the EDL and profoundly affect electrocatalytic processes, as evidenced by recent theoretical works^{11,12}.”

Reviewer 2:

Reviewer's Comments

In this manuscript, Prof. Lu and co-workers performed rigorous in situ SEIRAS studies under different potentials and CO pressures. Their main argument is the presence of two kinetically stable EDL structures and their effects on CO coverage and subsequent CO reduction activity.

This reviewer enjoyed reading the manuscript and would like to congratulate the authors for their impressive work. However, please note the following comments, one of which may be critical and needs to be clarified before its publication.

Response:

We thank the reviewer for the positive assessment.

A critical issue raised by this reviewer was found in the experimental section. The authors prepared an electrode with the Nafion ionomer to immobilize Cu powder on the Si ATR. Before realizing the use of the Nafion ionomer, this reviewer simply assumed an EDL structure relevant to the ideal GCS model, although the model was developed for electrolytes with low salt concentration. However, in the experiment, a thin layer of Nafion was found between the Cu and electrolyte interfaces, which is different from the assumed GCS model. The presence of the Nafion was also omitted in the schematic representation of Figure 2. In the literature, the importance of the electrode-ionomer interface in electrocatalysis has been suggested and its different EDL properties have also been presented. In the performance analysis, bare Cu without ionomer was seemingly used. Therefore, to clarify the authors' argument, it may be necessary to perform identical experiments with Cu electrode on ATR without Nafion ionomer.

Response:

We appreciate the reviewer's comment on the potential impact of using Nafion ionomer in our electrode preparation, which might lead to a potential deviation from the GCS model. In response to this concern, we conducted additional SEIRAS investigations on electrodeposited Cu films to rule out the effect of the Nafion ionomer on the spectroscopic results in this work. SEIRA spectra collected on electrodeposited Cu films (Figure R5) are consistent with those with the polycrystalline Cu powder

electrode with Nafion ionomer (Fig. 1). As shown in Figure R5, when CO was pressurized at -0.9 V from 1 atm to 40 barg, the CO_L peak intensity obtained on the electrodeposited Cu film barely changed (Figure R5a). However, when CO was pressurized at 0 V before shifting the electrode potential to -0.9 V, there was a noticeable increase in CO_L peak intensity, directly correlating with the CO pressure (Figure R5b). In addition, the potential dependent CO adsorption on electrodeposited Cu film shows consistent behavior as on the polycrystalline Cu powder electrode, which is discussed in detail in our response to the next comment. These results suggest that the presence of the Nafion ionomer does not alter the spectroscopic observations, supporting the validity of our model.

Figure R5. Pressure dependent CO_L band on electrodeposited Cu film electrode at -0.9 V when the CO pressure is elevated at (a) -0.9 V and (b) 0 V in 0.1 M potassium phosphate buffer electrolyte (pH=8).

Action:

We added the following sentence to the “Electrode Preparation” section on page 11 “To exclude the potential influence of Nafion ionomer to the CO adsorption behavior, electrodeposited Cu films on Au coated silicon ATR crystals were also prepared. Cu film electrodeposition was conducted at constant current of -260 $\mu\text{A cm}^{-2}$ for 8 min in an Ar-saturated solution containing 5 mM CuSO₄ and 50 mM H₂SO₄ following previous works^{34,60}. Both Cu powder and Cu film electrodes exhibited consistent CO adsorption behavior at elevated CO pressures (Fig. 1 and Supplementary Fig. 9),

suggesting that Nafion has a minimal impact on CO adsorption under our specified electrochemical conditions.”

Figure R5 is added to the SI as Supplementary Fig. 9.

Supplementary Fig. 9. Pressure dependent CO_L band on electrodeposited Cu film electrode at -0.9 V when the CO pressure is elevated at (a) -0.9 V and (b) 0 V in 0.1 M potassium phosphate buffer electrolyte (pH=8).

The following comments are my personal questions or some minor questions.

In situ SEIRAS analysis during potential swing from -0.9 to 0 and back to -0.9, and that with cation chelating agent or organic cation strongly support the authors' arguments. Another effective strategy to control the EDL structure will be to control the pH of the electrolyte, which changes the potential difference apart from the PZC of the Cu electrode. Could the authors show how the electrolyte pH affects the CO coverage or SEIRAS signals during the CO reduction reaction?

Response:

We appreciate the reviewer's insightful question regarding the impact of potential difference apart from the PZC on the EDL structure and the corresponding influence on CO adsorption. We recognize that variations in electrolyte pH can shift the PZC of the Cu electrode, leading to changes in the potential difference across the EDL determined by the difference between the PZC and the applied electrode potential. Auer et al.

reported that the PZFC of Cu(111) shifted by ~ 88 mV/pH in alkaline electrolytes (*J. Phys. Chem. C* 2021, 125, 9, 5020–5028), while Sobkowski and coworkers showed that the pH dependencies of PZC for all low index single crystal Cu facets and polycrystalline Cu were aligned in weakly acidic electrolytes (*J. Electroanal. Chem.* 2004, 567, 95–102). Thus, the potential drop across the EDL at the same SHE potential could be tuned by varying the electrolyte pH. Since the rate of PZC change with the electrolyte pH varies with the electrolyte pH and the crystallographic facet of Cu, there is a significant degree of uncertainty in the numerical values of PZC shifts with the electrolyte pH. Fortunately, the general direction of the PZC with pH is consistent, i.e., PZC on Cu surfaces becomes more negative with increasing electrolyte pH, enabling a qualitative analysis.

In response to the reviewer's comment, we performed potential-dependent SEIRAS experiments in electrolytes of pH 8 and pH 13.3 on **electrodeposited Cu films** to avoid the potential influence of Nafion ionomer, while maintaining an identical potassium concentration in the electrolyte. The spectra were compared on the SHE potential scale. As shown in Figure R6, the spectra in electrolyte of pH=8 (Figure R6a, c) behaved consistently with those obtained on polycrystalline Cu powder (Fig. 3), consistent with the results in the response to first comment from Review #2. We assume the rate of PZC shift in the -88 mV/pH reported by Auer et al. (*J. Phys. Chem. C* 2021, 125, 9, 5020–5028) in the following analysis because this value was determined in a similar pH range. When the electrolyte pH was shifted from 8 to 13.3, the PZC would cathodically shift by 446 mV. The potential at which EDL_{hp} and EDL_{lp} were established (-1.37 V) was more negative than the PZC in either electrolyte, as the PZC was determined to be ~ 0.7 V on Cu(111) at pH=13 (*J. Phys. Chem. C* 2021, 125, 9, 5020–5028). Thus, increasing the electrolyte pH from 8 to 13.3 reduces the potential drop across the EDL. This situation is similar to shifting the electrode potential from $-0.9 V_{\text{RHE}}$ to $-0.6 V_{\text{RHE}}$ in Figure R3. The further away the electrode potential is from the PZC, the more pronounced is the difference between EDL_{hp} and EDL_{lp} is expected. This is indeed the case. The relative difference of the CO_L band area at the same potential in the cathodic and anodic scans is significantly greater at pH = 8 than that at pH = 13.3 (Figure R6), because the potential drop across the EDL is greater in the former case. The CO_L peak was notably broadened at pH = 13.3, which was attributed a less homogeneous Cu surface in alkaline conditions (*Nat. Commun.* 2021, 12, 3264). Experiments were not conducted at lower electrolyte pHs because vigorous bubble produced by hydrogen evolution reaction disrupted the physical structure of the EDL and jeopardized the stability of Cu/Au film in acidic environment. Thus, shifting the potential drop across the EDL by varying the electrolyte pH yielded a qualitative consistent trend with changing the electrode potential, suggesting the reliability of the mechanism proposed

in this work.

We value the reviewer's insightful question and its relevance to our research. However, integrating additional experiments or detailed discussions into the current manuscript might distract from its integrity and cohesion. We believe pH impact on EDL structure and CO coverage presents an interesting direction for future work, deserving separate and in-depth exploration.

Figure R6. Potential dependent CO_L band on electrodeposited Cu film electrode when potential stepping anodically after CO is pressurized to 30 barg at -1.37 V vs. SHE (red lines) and then stepping cathodically (blue lines) in (a) phosphate buffer electrolyte of pH=8 and (b) potassium hydroxide electrolyte of pH=13.3 with the same potassium concentration.

Action:

We added on page 10, at the end of the conclusion section: “It is crucial to recognize that the rigidity of EDL structure is intimately associated with the electrostatic forces within it, which are notably affected by the pH of the electrolyte^{56,57}. This specific interaction between pH and EDL structure presents an intriguing area for future research.”

In the EDL_{ip} and EDL_{hp} conditions, different production rates of CO reduction are shown in Fig. 1c. However, these differences seem to be limited to H₂ and ethylene production rates and not to other products. If CO coverage determines the overall production rates, why are those of other products almost unaffected?

Response:

We would like to clarify that while the differences in Faradaic efficiency are more pronounced for H₂ and ethylene than that for other products, the partial current densities (i.e., reaction rates) of other CO reduction products do decrease when CO is pressurized at -0.9 V than those when CO is pressurized at 0 V (Figure R7), resulting from the overall decrease in CORR current density under the former condition (Fig. 1c). The extent of this decrease in partial current density towards a specific product is related to the dependence of its formation pathway on CO coverage. Since the yield or FE of specific product at different conditions depends sensitively on the reaction pathway and the associated rate-determining step, it remains unclear how the EDL_{ip} and EDL_{hp} conditions impact the rate of individual pathways. This could be a fruitful direction for a future study.

Figure R7. Comparison of the partial current densities at p_{CO} of 30 barg on polycrystalline Cu foil when the CO pressure is elevated at 0 V and -0.9 V in 0.1 M potassium phosphate buffer electrolyte (pH=8).

The authors measured the double layer capacitance simply by dividing the released charge by the voltage pulse. However, this may not be a general method to estimate EDL capacitance, but EIS method has been commonly used. If the capacitance is estimated by the EIS method, will the authors get identical results?

Response:

We thank reviewer's suggestion. In response, we performed EIS experiments to measure the EDL capacitances under two different conditions. The electrochemical cell was first pressurized with CO at either -0.3 V or 0 V to 30 barg and allowed to stabilize for 15 min to reach the dissolution and adsorption equilibria. Subsequently, the electrochemical impedance measurements were conducted at -0.3 V using a Gamry Reference 600+ Potentiostat within the frequency range between 30 kHz and 1 Hz and 10 mV amplitude. The obtained Nyquist plots were fitted by Zview software with an equivalent circuit of R_s - R_{ct} /CPE (Figure R8), where R_s is the solution resistance and R_{ct} is the charge transfer resistance. The capacitive effects of EDL were modeled using a constant phase element (CPE) to offset the influence of the distributed time constant. The impedance of CPE is $1/Y_0(j\omega)^n$, where Y_0 is the CPE coefficient and the exponent, n , is in the range of $0 \leq n \leq 1$ (*Nat. Rev. Methods Primers* 2021, 1, 41; *Langmuir* 2020, 36, 4250–4260). The equivalent capacitance was calculated using the method reported by Brug et al. with eq (1) (*J. Electroanal. Chem. Interfacial Electrochem.* 1984, 176, 275-295; *Electrochim. Acta* 2010, 55, 6218-6227).

$$C_{eff} = Y_0^{\frac{1}{n}} \cdot \left(\frac{1}{R_u} + \frac{1}{R_{ct}} \right)^{\frac{n-1}{n}} \quad (1)$$

The specific C_{EDL} was then determined by normalizing C_{eff} to the geometric surface area of Cu foil, which was determined to be 2.2 cm². As is shown in Figure R8, the EDL capacitances when CO was pressurized at -0.3 V and 0 V determined with EIS method were 57.9 and 17.3 $\mu\text{F cm}^{-2}$, respectively. These values align closely with the capacitances determined by analyzing transient current response to voltage pulse, with only minor discrepancies within 2 $\mu\text{F cm}^{-2}$. Both methods consistently indicate that EDL_{hp} exhibits a much smaller specific capacitance compared to that of the EDL_{lp}.

Figure R8. Nyquist plot at -0.3 V when CO was pressurized to 30 barg at -0.3 V and 0 V in 0.1 M potassium phosphate buffer electrolyte (pH=8) in electrochemical impedance measurement.

Action:

We revised the first sentence in “Double layer capacitance measurement” section on page 13 into “The electric double layer capacitances of EDL_{lp} and EDL_{hp} were measured by potential step voltammetry analyzing transient current response to a voltage pulse and electrochemical impedance spectroscopy (EIS) at -0.3 V without CORR^{62,63}.” We included the EIS data and its fitting results in Supplementary Fig. 5 and Supplementary Table 1. We also added the following sentences to the end of this paragraph “In EIS measurement, the electrochemical cell was first pressurized with CO at either -0.3 V or 0 V to 30 barg and allowed to stabilize for 15 min to reach the dissolution and adsorption equilibria. Subsequently, the EIS measurements were conducted at -0.3 V using a Gamry Reference 600+ Potentiostat within the frequency range between 30 kHz and 1 Hz and 10 mV amplitude. The obtained Nyquist plots were fitted by Zview software with an equivalent circuit of R_s - R_{ct} /CPE (Supplementary Fig. 5, Supplementary Table 1), where R_s is the solution resistance and R_{ct} is the charge transfer resistance. The capacitive effects of EDL were modeled using a constant phase

element (CPE) to offset the influence of the distributed time constant. The impedance of CPE is $1/Y_0(j\omega)^n$, where Y_0 is the CPE coefficient and the exponent, n , is in the range of $0 \leq n \leq 1$ ^{15,63}. The equivalent capacitance was calculated using the method reported by Brug et al. with eq (2)⁶⁴.

$$C_{eff} = Y_0^{\frac{1}{n}} \cdot \left(\frac{1}{R_u} + \frac{1}{R_{ct}} \right)^{\frac{n-1}{n}} \quad (2)$$

The specific C_{EDL} was then determined by normalizing C_{eff} to Cu foil's geometric surface area of 2.2 cm^2 .

Supplementary Fig. 5. Comparison of specific double layer capacitance at -0.3 V and p_{CO} of 30 barg on polycrystalline Cu foil when CO is pressurized at -0.3 V and 0 V in 0.1 M potassium phosphate buffer electrolyte of $\text{pH}=8$ using (a) potential step voltammetry and (b) electrochemical impedance spectroscopy.

Supplementary Table 1. Simulated impedance parameters of Nyquist plots at -0.3 V in 0.1 M potassium phosphate buffer electrolyte ($\text{pH}=8$) when CO was pressurized at -0.3 V and 0 V . The geometric surface area of Cu foil electrode was 2.2 cm^2 .

Condition	R_u /(Ω)	R_{ct} /(Ω)	Y_0 /($\mu\Omega^{-1}\cdot\text{s}^n$)	n	C_{eff} /(μF)	C_{EDL} /($\mu\text{F}\cdot\text{cm}^{-2}$)
CO pressurized at -0.3 V	9.3	8875	263.5	0.89	127.4	57.9
CO pressurized at 0 V	9.6	9393	83.3	0.90	38.1	17.3

Some typos (e.g. the 4th line of the second paragraph of the introduction) should be corrected.

Response:

We thank the reviewer for pointing out the error in typos. We have corrected these errors in the revised manuscript.

REVIEWERS' COMMENTS

Reviewer #1 (Remarks to the Author):

I have carefully read Lu's Response to the Reviewer. Lu and coworkers have already given a strong explanation of the electron microscopy results for the possible structural changes in the catalyst caused by the order of applied potentials and pressure. Possible controversies over the use of different materials for testing and characterizing are also explained. They explain the significance of the mechanistic studies for the application of CO(2)RR and the fundamental understanding of double layer capacitance. Spectroscopic and experimental explanations are also given for the ability of double electric layer capacitance to stabilize under high pressure and CORR instability conditions. Schematic diagrams of their custom-designed high pressure surface enhanced infrared absorption spectroelectrochemical cell are provided, as well as an illustration of the physical drawings. We therefore believe that this work demonstrates, through experimental evidence such as high-pressure in situ Raman characterization and CORR testing, that the applied potential and pressure sequence affects the rigidity of the double electric layer capacitance and thus alters CO adsorption, and two modeling explanations are clearly provided. All in all this work provides fundamental theoretical insights into CORR performance effects. Thus, we consider that the manuscript is suitable for publication on Nature Communications now.

Reviewer #2 (Remarks to the Author):

All concerns raised by this reviewer have been fully addressed in the revised manuscript. This reviewer is satisfied with their fruitful responses and supports its publication in the journal Nature Communications. This reviewer again congratulates the authors on their achievements.